

# Association of *IGF1* single-nucleotide polymorphisms with myopia in Chinese children

Tianyu Cheng[1,2,*], Jingjing Wang[1,*], Shuyu Xiong[1,2], Bo Zhang[1], Qiangqiang Li[3], Xun Xu[1,2] and Xiangui He[1,2]

[1] Department of Preventative Ophthalmology, Shanghai Eye Disease Prevention and Treatment Center, Shanghai Eye Hospital, Shanghai, China

[2] Department of Ophthalmology, Shanghai General Hospital, Shanghai Jiao Tong University, National Clinical Research Center for Eye Diseases, Shanghai Key Laboratory of Ocular Fundus Diseases, Shanghai Engineering Center for Visual Science and Photomedicine, Shanghai engineering center for precise diagnosis and treatment of eye diseases, Shanghai, China

[3] Baoshan Center for Disease Prevention and Control, Shanghai, China

[*] These authors contributed equally to this work.

Corresponding author
Xiangui He, xianhezi@163.com

## ABSTRACT

**Purpose**. To investigate the association between insulin-like growth factor 1 (*IGF1*) single-nucleotide polymorphisms (SNPs) and myopia in a young Chinese population.

**Methods**. A total of 654 Chinese children aged 6–13 years from one primary school participated in our study and underwent a series of comprehensive ocular examinations, including cycloplegic refraction and measurements of axial length. Myopia was defined as a spherical equivalence (SE) $\leq -0.5$ D in the worse eye. In total, six tagging SNPs of *IGF1* were genotyped using the PCR-LDR (Polymerase Chain Reaction-Ligation Detection Reaction) method. We tested four different genetic modes (the allele, dominant, recessive, and additive models) of these SNPs and used multivariate logistic regression to calculate the effect of SNPs on myopia. In addition, we conducted a haplotype analysis with a variable-sized slide-window strategy.

**Results**. Overall, 281 myopic children and 373 non-myopic controls were included in the analysis. The SNP rs2162679 showed a statistical difference between the two groups in both the allele ($p = 0.0474$) and additive ($p = 0.0497$) models. After adjusting for age and gender, children with the genotype AA in the SNP rs2162679 had a higher risk of myopia than those with the genotype GG (OR = 2.219, 95% CI [1.218–4.039], $p = 0.009$). All haplotypes that varied significantly between the two groups contained the SNP rs2162679, and the four-SNP window rs5742653–rs2162679 had the lowest $p$ value (Chi square = 5.768, $p = 0.0163$). However, after permutation tests, none of the associations remained statistically significant.

**Conclusion**. The SNP rs2162679 in *IGF1* was associated with myopia in a young Chinese population. The G allele in the SNP rs2162679 may protect against myopia.

## INTRODUCTION

Myopia is the most common refractive disorder worldwide, especially in Asian countries, such as China, Japan, and Singapore, and its prevalence is rising (*Ang et al., 2019*; *Morgan, Ohno-Matsui & Saw, 2012*). *Holden et al. (2016)* predicted that by the year 2050, almost half of the global population would be myopic, with 10% progressing to high myopia. High myopia could be closely associated with complications like glaucoma, cataracts, retinal detachment, macular hemorrhage, and choroidal neovascularization (*Ikuno, 2017*), which is one of the leading causes of blindness and results in financial burden (*Iwase et al., 2006*; *Xu et al., 2006*).

Myopia is a complex disease affected by both genetic and environmental factors, whose etiology and pathogenesis remain unclear (*Morgan et al., 2018*). Previous studies have identified 26 myopia-related loci (*MYP1–MYP26*) through various methods, such as family-based linkage analyses, genome-wide association studies, and whole exon sequencing studies (*Cui et al., 2017*; *Guo et al., 2015*; *Wang et al., 2017*; *Xiao et al., 2016*). Some candidate genes are associated with high myopia, while some are related to moderate myopia (*Zhang, 2015*). However, the genetic effect on myopia is an interactive process involving multi-genes and is heterogeneous among different populations.

In the *MYP3* locus on the chromosome 12q23.2, the insulin-like growth factor 1 (*IGF1*) gene codes for IGF-1, which is similar to insulin in structure and takes part in many physiological process including aging, apoptosis, development, cellular growth, metabolism, protein translation, and differentiation (*D'Mello et al., 1993*; *Werner & Le Roith, 2000*). IGF-1 is a kind of single-chain peptide composed of 70 amino acids which was also known as somatomedin C (*Savage, 2013*). It plays an important role in human growth and development through GH-IGF-1 axis (growth hormone) (*Furlanetto & Cara, 1986*). IGF-1 is found to be expressed in most of the tissues and the locally produced IGF-1 may be more important than its circulating endocrine form produced by liver (*Han et al., 1988*; *Le Roith et al., 2001*; *Yakar et al., 1999*). During puberty and pre-puberty, which are critical times for refraction development, IGF-1 may also be the inner motivation for ocular growth. For instance, after an intravitreal injection of IGF-1, the axial length of chicks grew longer (*Zhu & Wallman, 2009*). In addition, the axial length of patients with Laron syndrome, a disease that causes congenital growth hormone resistance, became normal, to a degree, after IGF-1 supplementation (*Bourla et al., 2006*). These studies have suggested that IGF-1 could contribute to ocular growth and the progression of myopia.

Genetic studies have been used to genotype the single-nucleotide polymorphisms of *IGF1* and have shown that it is significantly associated with high or extreme myopia in Caucasian and Chinese populations (*Mak et al., 2012*; *Zhuang et al., 2012*). However, these associations were not replicated in a Polish family cohort and in two Japanese studies (*Miyake et al., 2013*; *Rydzanicz et al., 2011*; *Yoshida et al., 2013*). Nevertheless, all previous studies focused on the relationship between single-nucleotide polymorphisms (SNPs) in *IGF1* and high myopia, whilst the association of SNP in *IGF1* with mild or moderate myopia, which usually occurs during puberty and pre-puberty, remains unknown. Here,

 

we aimed to show that genetic variants of *IGF1* contributed to the development of myopia in a young Chinese Han population.

## METHODS

### Participants

We used random cluster sampling to recruit a study population of children in grades 1–6 of a primary school in Baoshan District, Shanghai, China. All participants underwent comprehensive ophthalmologic tests, including an ophthalmologist exam and measurements of axial length and spherical power. All participants were divided into two groups according to their spherical equivalence (SE), which was measured using an automated computer refractometer after inducing cycloplegia (model: KR-8900; Topcon, Tokyo, Japan).

We induced cycloplegia by administering one drop of topical 0.5% proparacaine (Alcaine; Alcon, USA), followed by two drops of 1% cyclopentolate (Cyclogyl; Alcon, Fort Worth, TX, USA) in each eye, with a 5-minute interval between each drop. Pupil size and pupillary light reflex were examined at least 30 min after the last drop of cyclopentolate. Cycloplegia was defined as the absence of a pupillary light reflex and a pupil size larger than 6 mm. We measured axial length using an IOL-master (version 5.02; Carl Zeiss Meditec, Oberkochen, Germany). To avoid the invasiveness of blood testing, we collected saliva to extract DNA and target SNP genotyping. We calculated SE using the following formula: SE = sphere + 0.5 * cylinder. We diagnosed myopia as SE $\leq -0.50$ D in the worse eye. Anyone who had amblyopia, glaucoma, cataracts, or other genetic diseases associated with myopia were excluded from our study.

This study was approved by the Institutional Review Board of Shanghai General Hospital, Shanghai Jiaotong University and was performed in accordance with the principles of the Declaration of Helsinki. Informed consent was obtained from all participants and their guardians.

### SNP selection and genotyping

To select tagSNPs from *IGF1* genes, we used Haploview software version 4.2 (https://www.broadinstitute.org/haploview/haploview) on the Han Chinese in Beijing (CHB) dataset from the 1000 Genomes Project (http://www.internationalgenome.org). Inclusion criteria were as follows: minor allele frequency (MAF) >0.1 and pairwise $r^2 > 0.8$. In total, six SNPs were used in the analysis, including rs6214, rs5742653, rs4764697, rs12423791, rs2162679, and rs5742612. Among these tagSNPs, rs6214 and rs12423791 were associated with high myopia in previous studies.

We extracted DNA with a salivary genomic DNA rapid extraction kit (KL3216T-50, Shanghai Ze Ye Biological Technology Company, China) from oral epithelial cells under standard conditions. We performed genotyping for the selected tagSNPs using the PCR-LDR method (Polymerase Chain Reaction-Ligation Detection Reaction) with technical support from the Shanghai Biowing Applied Biotechnology Company. We selected 5% of samples for duplicate detection through simple random sampling.

**Table 1  Primer Sequences and PCR Length.**

| SNPs | Upper primer sequence (5′–3′) | Lower Primer sequence (5′–3′) | PCR length |
|------|-------------------------------|-------------------------------|------------|
| rs6214 | TTCCCTCTCAACAAAACTTT | GCTTTCCTCCTTGGGGGATT | 147 |
| rs5742653 | CAGCCTGAGCAGCATAGTGA | GCTCACTACAGCGTCAACCT | 144 |
| rs4764697 | TGGTCCCTGGATGTTGTTTAG | CCTCATCGCCCACCAAAACT | 151 |
| rs12423791 | TAGGCCCCATCTCTTTGCTG | TTGCTGCCTCCTGTTCACAT | 128 |
| rs2162679 | TTGAACAGGAAAACCCCACT | ACTGCATTTTTCTCAACAAG | 147 |
| rs5742612 | CTTTGCCTCATCGCAGGAGA | ATTGGAAGACAGCACTCGGG | 149 |

PCR-LDR method combines PCR and LDR together and could detect variations in SNPs by Taq ligase. The primer sequences of these six tagSNPs for PCR were shown in Table 1 and the corresponding probe sequences for LDR were shown in Table 2. PCR was conducted for the amplification of selected SNP sites. The PCR reaction system (20 µl) contained 2 µl 1 × buffer, 0.6 µl Mg2+ (3 mmol/L), 2 µl dNTPs (2 mmol/L each), 0.2 µl Taq polymerase (1 U), 4 µl 1 × Q-solution, 0.4 µl primer mix, 9.8 µul ddH2O and 1ml genomic DNA of samples. The PCR program on MJ PTC-200 (MJ Research, USA) was as followed: denaturing at 95 °C for 2 min; then denaturing at 94 °C for 30 s, annealing at 56 °C for 90 s, extension at 65 °C for 30 s, repeat 40 cycles; afterwards, extension for 10 min at 65 °C. Then multiplex LDR was performed for SNPs detection. The LDR reaction system (10 µl) consist of 1 µl 1 × buffer, 1 µl probe mix, 0.05 µl Taq DNA ligase (2 U), 3.95 ul ddH2O and 4 µl amplification products. The LDR program on Gene Amp PCR system (model 9600, Norwalk, CT.06859 USA) was set as followed: initial denaturing at 95 °C for 2 min, followed by 40 cycles of denaturing at 94 °C for 15 s, and annealing at 50 °C for 25 s. At last, Genemapper (ABI, Inc.) was applied for data analysis and genotyping.

## Statistical analysis

We analyzed clinical data using Statistical Product and Service Solutions (SPSS version 25, IBM, USA) and genetic data using Haploview version 4.2 and gPLINK version 1.07. First, we ensured that each SNP of the myopia and non-myopia groups passed the Hardy-Weinberg equilibrium (HWE) test. We used a Chi Square test of four different genetic models (the allele, dominant, recessive, and additive models) to show the distributions of different alleles and genotypes as well as their potential genetic modes. While considering confounding factors, we conducted a multivariate logistic regression analysis adjusting for age and gender. In addition, we analyzed the associations between different haplotypes and myopia using a variable-sized slide-window strategy in Haploview software. Significance was set at $p < 0.05$, and we used the 10,000 times permutation test for multiple comparisons. The power of test was calculated using two independent proportions power analysis (PASS version 11.0).

## RESULTS

There were 281 myopes and 373 non-myopes with mean ages of $9.84 \pm 1.55$ years and $8.06 \pm 1.43$ years, respectively. There were no significant differences in gender between the groups. The mean SE of participants' right eye was $-2.55 \pm 1.64$ D in the myopia

**Table 2  Probe Sequences and LDR Length.**

| Probe name | Probe sequence (5′–3′) | LDR length |
|---|---|---|
| rs6214_modify | P-GTTAAGTCTGCAGAAGACTGTTTTTTTTTTTTTTTTTTTTTTTTTTTTTTTTTTTTTTT-FAM | |
| rs6214_A | TTTTTTTTTTTTTTTTTTTTTTTTTTTTTTTTTTTTTCATCTAACTATGACAGAAAACAT | 117 |
| rs6214_G | TTTTTTTTTTTTTTTTTTTTTTTTTTTTTTTTTTTTTCATCTAACTATGACAGAAAACAC | 119 |
| rs5742653_modify | P-TGGGACTGCAGGCATGCATTTTTTTTTTTTTTTTTTTTTTTTTTTTTT-FAM | |
| rs5742653_A | TTTTTTTTTTTTTTTTTTTTTTTTTTTCCCACCTCAGCCTTCTGAGTAGT | 97 |
| rs5742653_G | TTTTTTTTTTTTTTTTTTTTTTTTTTTCCCACCTCAGCCTTCTGAGTAGC | 99 |
| rs4764697_modify | P-TGTGCAAGACTGCTTGAGGCTTTTTTTTTTTTTTTTTTTTTTTTTTTTTTTT-FAM | |
| rs4764697_C | TTTTTTTTTTTTTTTTTTTTTTTTTTTTTCAGCCTCCTCCATGATCGTGCTG | 105 |
| rs4764697_T | TTTTTTTTTTTTTTTTTTTTTTTTTTTTTCAGCCTCCTCCATGATCGTGCTA | 107 |
| rs12423791_modify | P-TCAGTATCACTATTTCCTCTTTTTTTTTTTTTTTTTTTTTTTTT-FAM | |
| rs12423791_C | TTTTTTTTTTTTTTTTTTTTTAATGTATCTTCAGAATGCTCAG | 89 |
| rs12423791_G | TTTTTTTTTTTTTTTTTTTTTTTAATGTATCTTCAGAATGCTCAC | 91 |
| rs2162679_modify | P-CTACATAGCCCAAAACACTGTTTTTTTTTTTTTTTTTTTTT-FAM | |
| rs2162679_A | TTTTTTTTTTTTTTTTTTTCCGCATGGAAATCTTCCACCCT | 81 |
| rs2162679_G | TTTTTTTTTTTTTTTTTTTTTCCGCATGGAAATCTTCCACCCC | 83 |
| rs5742612_modify | P-GGTGTGATCTCATTTCCTAGTTTTTTTTTTTTTTTTTTTTTTTTTTTTTTT-FAM | |
| rs5742612_C | TTTTTTTTTTTTTTTTTTTTTTTTTTTTTTTCTTGTCCCAGTTGCCAAGTGAGG | 101 |
| rs5742612_T | TTTTTTTTTTTTTTTTTTTTTTTTTTTTTTTCTTGTCCCAGTTGCCAAGTGAGA | 103 |

**Table 3  Characteristics of Study Participants.**

| | | | Myopes (n = 281) | Non-myopes (n = 373) | p |
|---|---|---|---|---|---|
| Age (years) | | | 9.84 ± 1.55 | 8.06 ± 1.43 | <0.001 |
| Males (%) | | | 159 (56.6%) | 193 (51.3%) | 0.182 |
| AL | OD[a] | | 24.58 ± 1.04 | 23.02 ± 0.79 | <0.001 |
| (mm) | OS[b] | | 24.40 ± 1.81 | 23.00 ± 0.80 | <0.001 |
| MSE[c] | OD | | −2.55 ± 1.64 | 0.84 ± 0.81 | <0.001 |
| (D) | OS | | −2.25 ± 1.84 | 0.88 ± 0.83 | <0.001 |

**Notes.**
[a] OD, right eye.
[b] OS, left eye.
[c] MSE, mean spherical equivalence.

group and $0.84 \pm 0.81$ D in the non-myopia group ($p < 0.001$). The mean axial length of participants' right eye was $24.58 \pm 1.04$ mm for the myopes and $23.02 \pm 0.79$ mm for the non-myopes ($p < 0.001$). No significant difference was observed in SE and axial length between right and left eyes (Table 3).

All six tagSNPs were successfully genotyped, and all of the duplicated samples showed the same genotypes with their corresponding ones. The genotype distributions were all in line with HWE in both the myopia and non-myopia groups ($p > 0.05$). Their alleles and frequencies have been shown in Table 4.

We compared differences in genotype between the two groups using four genetic models: the allele, dominant, recessive, and additive models (Table 5). The SNP rs2162679 showed a

**Table 4  Genotypes and Hardy–Weinberg Equilibrium $p$ Values in Myopic and Non-Myopic Children.**

| SNP | Chromosome | Position | Alleles | MAF[a] | | HWE[b] $p$ value | |
|---|---|---|---|---|---|---|---|
| | | | | Myopia | Control | Myopia | Control |
| rs6214 | 12 | 102399791 | G > A | 0.459 | 0.488 | 0.572 | 0.051 |
| rs5742653 | 12 | 102442081 | A > G | 0.471 | 0.472 | 0.068 | 0.579 |
| rs4764697 | 12 | 102459394 | C > T | 0.180 | 0.164 | 0.546 | 1 |
| rs12423791 | 12 | 102465050 | G > C | 0.241 | 0.267 | 0.590 | 1 |
| rs2162679 | 12 | 102477481 | A > G | 0.321 | 0.374 | 0.848 | 0.787 |
| rs5742612 | 12 | 102481086 | T > C | 0.273 | 0.302 | 1 | 0.062 |

Notes.
[a]MAF, minor allele frequency.
[b]HWE, Hardy–Weinberg Equilibrium.

**Table 5  Association Tests of Six SNPs Between Myopes and Non-Myopes.**

| SNP | Model | | | | Corrected $p$ value[a] |
|---|---|---|---|---|---|
| | Allele | Dominant | Recessive | Additive | |
| rs6214 | 0.2981 | 0.174 | 0.7255 | 0.2782 | 0.7569 |
| rs5742653 | 0.9858 | 0.5636 | 0.5094 | 0.9853 | 1.0000 |
| rs4764697 | 0.4496 | 0.361 | 0.8901 | 0.4441 | 0.9100 |
| rs12423791 | 0.1464 | 0.266 | 0.1597 | 0.1436 | 0.4587 |
| rs2162679 | 0.0474[*] | 0.07926 | 0.159 | 0.04973[*] | 0.1761 |
| rs5742612 | 0.2539 | 0.1089 | 0.7883 | 0.2396 | 0.6886 |

Notes.
[*]Chi Square test $p < 0.05$.
[a]Multiple comparison corrected using the 10,000 times permutation test.

statistically significant association with myopia in the allele and additive models ($p = 0.0474$ and 0.04973, respectively). However, these associations did not exist after we conducted the 10,000 times permutation test. All the other SNPs were not significantly associated with myopia in these four models. After adjusting for age and gender, participants with the genotype AA in the SNP rs2162679 had a higher risk of myopia than those with the genotype GG (OR = 2.219, 95% CI [1.218–4.039], $p = 0.009$; Table 6), with post hoc power of 66.2% . Also, the genotype CC in rs12423791 and CT in rs5742612 were associated with a lower risk of myopia (OR = 0.407 and 0.648, respectively; 95% CI [0.185–0.896] and 0.443–0.948, respectively; $p = 0.026$) after adjusting for age and gender.

We performed a haplotype analysis between the myopia and non-myopia groups with a variable-sized slide-window strategy in Haploview software version 4.2 (Table 7, Fig. 1). The four-SNP window rs5742653–rs2162679 had the lowest $p$ value (Chi square = 5.768, $p$ = 0.0163), and all SNP windows with $p$ values less than 0.05 included the SNP rs2162679. However, no significant differences existed after the permutation test.

## DISCUSSION

This study investigated the association between the SNP rs2162679 in the intron of *IGF1* and myopia in a young Chinese population during puberty and pre-puberty. Those with

**Table 6  Association Tests Adjusted by Age and Gender in the Additive Model.**

| SNP | Genotype | OR | 95% CI | P |
|---|---|---|---|---|
|  | AA | 0.784 | (0.461, 1.336) | 0.371 |
| rs6214 | AG | 0.841 | (0.547, 1.294) | 0.432 |
|  | GG | Reference | / | / |
|  | AA | 1.307 | (0.760, 2.247) | 0.333 |
| rs5742653 | AG | 1.294 | (0.802, 2.088) | 0.290 |
|  | GG | Reference | / | / |
|  | CC | 1.298 | (0.389, 4.331) | 0.671 |
| rs4764697 | CT | 1.456 | (0.425, 4.982) | 0.550 |
|  | TT | Reference | / | / |
|  | CC | 0.407 | (0.185, 0.896) | 0.026[*] |
| rs12423791 | CG | 0.775 | (0.529, 1.136) | 0.192 |
|  | GG | Reference | / | / |
|  | AA | 2.219 | (1.218, 4.039) | 0.009[*] |
| rs2162679 | AG | 1.529 | (0.877, 2.893) | 0.127 |
|  | GG | Reference | / | / |
|  | CC | 0.523 | (0.252, 1.089) | 0.083 |
| rs5742612 | CT | 0.648 | (0.443, 0.948) | 0.026[*] |
|  | TT | Reference | / | / |

**Notes.**
*Logistic regression: $p < 0.05$ after adjusting for age and gender.

the genotype AA in rs2162679 had a higher risk of myopia than those with the genotype GG (OR = 2.219). The A >G variation in this position may protect against myopia. In anther word, individuals with G allele in rs2162679 of *IGF1* have less chance of getting myopia. During puberty and pre-puberty, increasing levels of hormones, such as growth factor and IGF-1, accelerate ocular growth and the development of body stature. These genetic variants in *IGF1* may be related to the onset or early development of myopia in young populations.

We found no significant association between rs6214, rs12423791, and myopia in this young Chinese population. However, within the *MYP3* locus, these SNPs in *IGF1* were previously associated with high myopia, a finding that has been replicated by several studies. *Metlapally et al. (2010)* were the first to test the SNPs in *IGF1* and found that rs6214 was significantly related to any type of myopia, including high myopia, in Caucasian populations. However, this conclusion was not confirmed in a Polish family cohort or in two Chinese and two Japanese studies (*Miyake et al., 2013*; *Rydzanicz et al., 2011*; *Yoshida et al., 2013*). A 2017 meta-analysis including eight case-control studies published before June 2016 reported no significant association between high myopia and the SNP rs6214 in *IGF1* (*Zhang et al., 2017*). Moreover, Zhuang et al. found that rs12423791, another SNP in *IGF1*, was associated with extreme myopia rather than high myopia in a Chinese population and that the haplotype containing this SNP showed a significant association with extreme myopia (*Mak et al., 2012*; *Zhuang et al., 2012*). However, Yoshida et al., Miyake et al., and

**Table 7  Haplotype Analysis Between Myopia and Non-Myopia Groups.**

| Haplotypes | Blocks with the lowest p values | Chi Square value | p value | Corrected p value[a] |
|---|---|---|---|---|
| **Two-SNP window** | | | | |
| rs6214–rs5742653 | AA | 1.14 | 0.2857 | 0.5587 |
| rs5742653–rs4764697 | GT | 0.534 | 0.465 | 0.8216 |
| rs4764697–rs12423791 | TC | 2.003 | 0.157 | 0.3452 |
| rs12423791–rs2162679 | GA | 5.394 | 0.0202[*] | 0.0566 |
| rs2162679–rs5742612 | AT | 3.134 | 0.0767 | 0.1970 |
| **Three-SNP window** | | | | |
| rs6214–rs4764697 | AAC | 0.746 | 0.3878 | 0.8988 |
| rs5742653–rs12423791 | GTC | 1.866 | 0.1719 | 0.5487 |
| rs4764697–rs2162679 | CGG | 2.137 | 0.1438 | 0.4215 |
| rs12423791–rs5742612 | GAT | 4.549 | 0.0329[*] | 0.1536 |
| **Four-SNP window** | | | | |
| rs6214–rs12423791 | AGCG | 1.456 | 0.2276 | 0.7044 |
| rs5742653–rs2162679 | GCGA | 5.768 | 0.0163[*] | 0.0586 |
| rs4764697–rs5742612 | TGAT | 1.587 | 0.2077 | 0.7061 |
| **Five-SNP window** | | | | |
| rs6214–rs2162679 | GGCGA | 3.99 | 0.0458[*] | 0.1853 |
| rs5742653–rs5742612 | GCGAT | 4.003 | 0.0454[*] | 0.2245 |
| **Six-SNP window** | | | | |
| rs6214–rs5742612 | GGCGAT | 2.937 | 0.0866 | 0.3999 |

Notes.
[*]Chi Square Test: $p < 0.05$.
[a]Multiple comparison corrected using the 10,000 times permutation test.

a 2017 Chinese study failed to replicate these associations (*Miyake et al., 2013*; *Wang et al., 2017*; *Yoshida et al., 2013*). This may be due to ethnic differences or the severity of myopia.

Unlike the positive relationship found in our study, two previous Japanese studies found no significant associations between rs2162679 in *IGF1* and high or extreme myopia (*Miyake et al., 2013*; *Yoshida et al., 2013*). Participants in our study were children in the puberty or pre-puberty stages, and most suffered from mild or moderate myopia. These findings suggested that rs2162679 was only associated with mild and moderate myopia rather than high or extreme myopia. To date, six MYPs (*MYP6–10* and *MYP14*) have been discovered and mapped in populations with mild and moderate myopia (*Cui et al., 2017*). Baltimore et al. mapped a myopia-related region in chromosome 1p36 in an Ashkenazi Jewish population with a mean SE of $-3.46$ D (*Wojciechowski et al., 2006*). In addition, chromosome 22q12 was associated with moderate myopia in 44 large American Ashkenazi Jewish families (MSE $= -4.67$ D) (*Stambolian et al., 2004*). Even though myopia is a complex disease affected by environmental factors, and high myopia is thought to be hereditary, it is necessary consider genetic factors in mild and moderate myopia.

Though intravitreal injections of insulin or IGF-1 in chicks results in extreme axial extension, changes in the axial length in mammals are relatively limited (*Feldkaemper,*
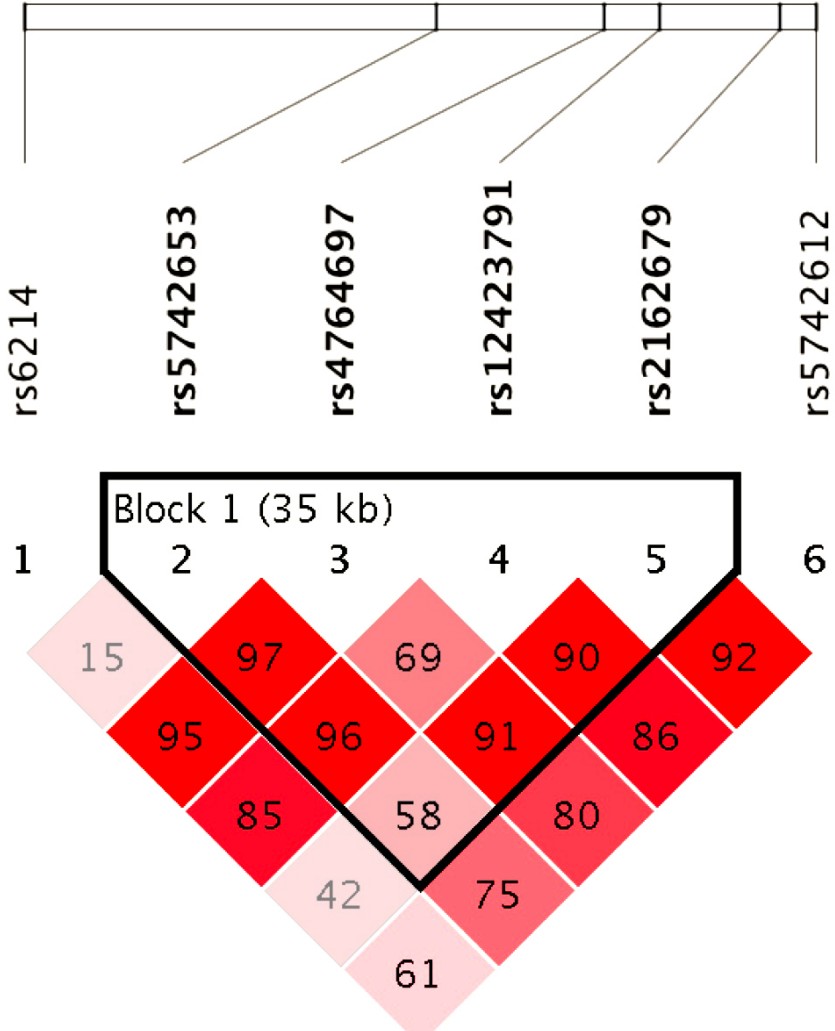

**Figure 1    Distribution of the tagging SNPs in IGF-1.**

*Neacsu & Schaeffel, 2009*; *Zhu & Wallman, 2009*). This may be due to variation in their ocular structures and metabolism pathways (*Mathis & Schaeffel, 2007*). There is cartilage in the sclera of chick eyes, which could be stimulated by IGF-1 via cartilage anabolism (*Martel-Pelletier et al., 1998*; *Rada, Thoft & Hassell, 1991*). In addition, this structure could support axial elongation in chicks better than in mammals, whose eyes do not contain cartilage. This may explain why IGF-1 resulted in extreme increases in axial length in chicks but has only been related to mild to moderate myopia in humans.

Interestingly, rs2162679 of *IGF1* had been reported to be associated with several kinds of cancer, in which the allele G could also decrease the risk of cancer onset (*Xu et al., 2019*). It reminded us that IGF-1 might play similar roles in myopia and cancer. More studies are needed to investigate the correlation between myopia and cancer.

However, this study had some limitations. First, the sample size was relatively small. Small sample sizes are more likely to result in false positives than larger sample sizes. They also have lower statistical power to detect positive results (*Yoshida et al., 2013*). Second, environmental factors, such as lack of outdoor activity or too much near work, are involved in the development of myopia, so we these may have confounded our analysis (*Saw et al., 2000*). Further studies with large sample sizes and in different populations are needed, especially those that stratify populations according to levels of myopia. Environmental factors and gene-environment interactions should also be taken into consideration. Last but not least, considering that rs2162679 locates in the intron region of *IGF1*, whether its variation could lead to protein function change remained unknown.

## CONCLUSION

We found that the SNP rs2162679 in *IGF1* was significantly associated with myopia in this young Chinese population, which provided clues for in-depth mechanism interpretation that IGF-1 may play a regulating role in the progression of myopia shift. Besides, it indicated that rs2162679 in *IGF1* could be one potential genetic loci for myopic risk prediction.

## ACKNOWLEDGEMENTS

We would like to thank all the subjects participating in this research.

### Funding

This work was supported by the National Natural Science Foundation of China (No. 81402695), the National Key R&D Program of China (2016YFC0904800, 2019YFC0840607), and the National Science and Technology Major Project of China (2017ZX09304010). The funders had no role in study design, data collection and analysis, decision to publish, or preparation of the manuscript.

### Grant Disclosures

The following grant information was disclosed by the authors:
National Natural Science Foundation of China: 81402695.
National Key R&D Program of China: 2016YFC0904800, 2019YFC0840607.
National Science and Technology Major Project of China: 2017ZX09304010.

### Competing Interests

The authors declare there are no competing interests.

### Author Contributions

- Tianyu Cheng conceived and designed the experiments, performed the experiments, analyzed the data, prepared figures and/or tables, authored or reviewed drafts of the paper, and approved the final draft.

- Jingjing Wang performed the experiments, analyzed the data, authored or reviewed drafts of the paper, and approved the final draft.
- Shuyu Xiong analyzed the data, authored or reviewed drafts of the paper, and approved the final draft.
- Bo Zhang and Qiangqiang Li performed the experiments, authored or reviewed drafts of the paper, and approved the final draft.
- Xun Xu conceived and designed the experiments, authored or reviewed drafts of the paper, and approved the final draft.
- Xiangui He conceived and designed the experiments, performed the experiments, authored or reviewed drafts of the paper, and approved the final draft.

### Human Ethics

The following information was supplied relating to ethical approvals (i.e., approving body and any reference numbers):

This study was approved by the Institutional Review Board of Shanghai General Hospital, Shanghai Jiaotong University (Ethical Application Ref: 2015KY148).

### Data Availability

The data is available at Figshare: Cheng, Tianyu (2019): Database of an association study between SNPs in IGF-1 and myopia .sav. figshare. Dataset. DOI 10.6084/m9.figshare.9735719.v1.

### Supplemental Information

Supplemental information for this article can be found online at http://dx.doi.org/10.7717/peerj.8436#supplemental-information.

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
