# Peer review of "Association of IGF1 single-nucleotide polymorphisms with myopia in Chinese children"

_PeerJ, doi:10.7717/peerj.8436_

## Round 0.1 · original submission · Major Revisions

The concerns of the two reviewers should be clearly addressed before a final decision can be made on your manuscript.

Reviewer 1 ·

Basic reporting

English clear, introduction not described well, current literature not cited.

Experimental design

no comments

Validity of the findings

no comments

Additional comments

Cheng et al., found an association between IGF-1 and myopia in Chinese children.The authors showed association of myopia and SNP rs2162679 (G). The G allele of this mutation confers protection against myopia in chinese children.

1. Author should take into account sampling locations in china. Are these samples collected from single place or multiple place? If they are collected from different place, are there any difference?
2. Introduction should be elaborated with more citation. What is the role of IGF-1?
3. The authors checked LD between SNPs. It is earlier reported that rs2162679 plays a role in cancer, did authors check the correlation between cancer and myopia?
4. If the protective allele (G) is present, is that means that individual is protected from myopia? Or it is chances of getting myopia is less?
5. Additionally, if an individual born with allele G, but in later life it becomes A, does that mean that individual is at risk?
6. Method/Results section should be elaborated, not very clear.
7. Authors should provide human chromosomal location of IGF-1.
8. Did authors also check the ancestral form of SNP? Whether it is present in great apes, archaic genome?

Reviewer 2 ·

Basic reporting

1. The manuscript is mostly well-formatted according to the journal's guidelines. There are, however, minor errors: a) text is full width justify instead of left margin justify, b) In the reference section, some references do not have the full name of the journal, only the abbreviation.
2. The most significant results were obtained for SNP rs2162679. I think that in the discussion section it is worth to add information about its location in IGF-1 (intron/exon) and assessment of possible effects on the gene.
3. Line 153: In my opinion a nucleotide change (rs2162679) with such a high frequency (MAF = 0.374) cannot be called a mutation.
4. Line 29: I think that to describe this age group, the word "children" or "pupils" would be better than "students".
5. In line 59 there should be "locus on chromosome" insdead of "locus of the chromosome".

Experimental design

1. In my opinion, the description of some fragments of research methods is too general and does not allow a full assessment of the correctness of the used techniques. For example no reaction conditions and reagents for ligation detection reaction method were given.
2. The power of the sample size is not given.
3. The methods section stated that 5% of the samples were used in duplication. However, the results section is missing information on this topic.

Validity of the findings

no comment

Additional comments

A well and clearly written manuscript. However, I find missing in the discussion clearly emphasize why these results are relevant.

---

## Round 0.2 · accepted · Accept

Dear Dr Cheng

Thank you for submitting your revised manuscript titled " Association of IGF-1 single-nucleotide polymorphisms with myopia in Chinese children”. You have adequately addressed the concerns of both reviewers and the manuscript has been accepted for publication in PeerJ.

Ziarih Hawi

Reviewer 1 ·

Basic reporting

The revised version of manuscript has clear and understandable english.

Experimental design

Primary aim of the study is within the scope of the journal. Methods are described sufficiently in the revised manuscript.

Validity of the findings

Conclusions are well stated, which is linked to the original reach question.

Additional comments

Thank you authors for reviewing manuscript.